# FROM GALORE TO WELORE: HOW LOW-RANK WEIGHTS NON-UNIFORMLY EMERGE FROM LOW-RANK GRADIENTS

## ABSTRACT

Modern Large Language Models (LLMs) are composed of matrices with billions of elements, making their storage and processing quite demanding in terms of computational resources and memory usage. Being significantly large, such matrices can often be expressed in low-rank format with potential to relax resource requirements. Unlike prior works which focus on developing novel matrix decomposition algorithms, in this work we first study the *emergence of low-rank structures* across matrices within different layers of LLMs and establish a **consequential relationship** between the gradient dynamics and emerging low-rank expressiveness of matrices. Our findings reveal that different layers exhibit varying levels of converged low-rank structure, necessitating a non-uniform rank reduction across them to minimize performance drop due to compression. In view of that, we present *Weight Low-Rank Projection* (**WeLore**) that **unifies weight compression and memory-efficient fine-tuning as ONE**, in a *data-agnostic and one-shot way*. WeLore capitalizes the *heavy-tail distribution of singular values* to identify a suitable rank reduction ratio for matrices within LLMs. Going beyond only as a compression technique, WeLore categorizes weight matrices into Low-rank Components (LRCs) and Non-Low-rank Components (N-LRCs) based on their ability to express themselves as low-rank. Our gradient perspective and extensive experiments illustrate that *LRCs tend to have better finetuning capabilities* and can closely mimic (sometimes outperform) the training loss trajectory and performance of full-finetuning with notable memory and compute footprint reduction. For example, finetuning a 50% compressed LLaMa-2 7B model using only a fraction of parameters in LRCs (WeLore) can **outperform its full finetuning** with $\sim 3\times$ better throughput and $\sim 0.6\times$ GPU requirement.

## 1 INTRODUCTION

In the modern era of deep learning, observing low-rank structures across gigantic matrices is common. Over the decades, low-rank structures have been notably useful and ubiquitous across numerous applications, such as image and data compression (Lingala et al., 2011; Arif et al., 2019; Yu et al., 2014), deep neural network compression (Hsu et al., 2022; Kaushal et al., 2023; Li et al., 2023; Jaiswal et al., 2023a; Wang et al., 2023), and recently for fine-tuning large language models (LLMs) (Hu et al., 2021; Dettmers et al., 2024; Meng et al., 2024; Biderman et al., 2024; Lialin et al., 2023). The storage efficiency and fine-tuning memory footprints associated with the large matrices of LLMs are currently prohibitive to unlocking the full potential of lightweight domain-specific applications around them. For example, regular 16-bit fine-tuning of a LLaMA-65B parameter model requires more than 780 GB of GPU memory (Dettmers et al., 2024), and the VRAM consumption for training GPT-3 175B reaches 1.2TB (Meng et al., 2024).

In recent efforts to address storage demands and computational complexity linked to the large matrices of LLMs, several works have been exploring the low-rank characteristics associated with weights and gradients (Zhao et al., 2024; Hu et al., 2021; Hsu et al., 2022; Kaushal et al., 2023; Li et al., 2023; Wang et al., 2023; Meng et al., 2024; Wang et al., 2024). One primary limitation of the existing works is an under-explored assumption of the uniform existence of low-rank structures across the pre-trained weights, with a main focus on developing matrix factorization techniques for LLM compression.

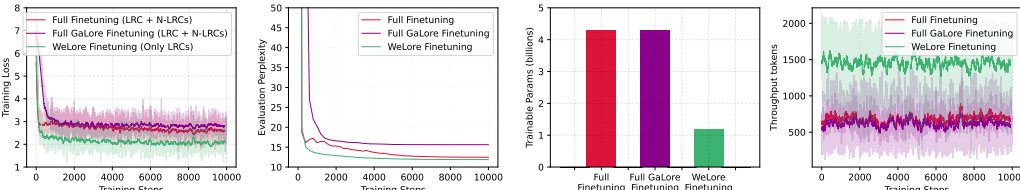

Figure 1: Continual-Finetuning statistics and performance comparison of a 50% low-rank compressed LLaMa-2 7B pretrained checkpoint from HuggingFace using C4 dataset. With exactly same hyper-paramter configurations, *WeLore can can **outperform** full-finetuning* with merely $\sim$**35%**$\times$ of trainable parameters while providing $\sim$**3**$\times$ **better throughput**.

Recently, (Sharma et al., 2023) interestingly found that it is often possible to significantly improve the performance of LLMs by selectively removing higher-order components of their weight matrices.

In this work, we first explore how the *low-rank structure emerges and differs* across weight matrices corresponding to different Attention and MLP layers within transformer blocks of LLMs. Motivated by the findings of Galore (Zhao et al., 2024), which establish that gradients during the pretraining of LLMs become low-rank, our work makes an effort to understand how the gradient behavior changes over time during LLM pretraining and attempts to establish a *consequential relationship* between the emergence of low-rank weight subspace and gradient subspace.

**Weight Low-Rank Subspace through the Lens of Gradient Behaviour:** Recently, GaLore (Zhao et al., 2024) theoretically argues that the gradient matrix becomes low-rank during training but does not establish *how the gradient behavior accumulates in the weight space*. Moreover, it provides no distinct consideration on training dynamics of different layers (*e.g.,* attention, MLP) across transformers blocks in LLMs. To this end, we first carefully investigated the gradient behavior of all weight matrices during back-propagation starting with random initialization (usually full-rank) during full pretraining. We found that gradient matrices of some layers (*e.g.,* majority of middle MLP matrices) saturate significantly within a short span of training iterations. On the other hand, gradients for some weight matrices (*e.g.,* attention matrices from the first and last transformer blocks) continuously carry rich error signals from training data and develop low-rank gradient subspace throughout the training. We conjecture that as a consequence of the cumulative accumulation of gradients within a low-rank gradient subspace, the corresponding weight matrices exhibit the emergence of high-quality stable low-rank subspace. Our study found that different layers within an LLM pose varying levels of converged low-rank structure, which should be accounted for during low-rank decomposition.

This new gradient perspective into nonuniform weight ranks unfolds several interesting dimensions:

- Weight matrices corresponding to different layers across transformer blocks can be broadly categorized as: ① ***Low-rank Components (LRCs)*** that exhibit high-quality low-rank structure (can be estimated by heavy-tail in sorted singular values obtained with SVD) and their gradients can carry rich error signals from data; ② ***Non-Low-rank Components (N-LRCs)*** with non-converged low-rank structure (missing heavy-tail in singular values distribution) and cannot be low-rank factorized without introducing noticeable reconstruction error.

- It provides us a unique opportunity to unify weight compression and memory-efficient fine-tuning (MEFT) **as ONE**: (a) **compression angle:** LRCs with stabilized low-rank weight structure can be factorized by SVD to significantly high compression ratio; and (b) **MEFT angle:** when fine-tuning, we back-propagate only over LRCs in their low-rank decomposed format to make the most effective gradient progress while leaving N-LRCs frozen.

Our aforementioned discussion led to **Weight Low-Rank Projection (WeLore)**, an *one-shot and data-agnostic layer-wise non-uniform* rank reduction technique based on the emerged low-rank subspaces in LRCs and N-LRCs. More specifically, to achieve a target rank reduction ratio, we exploit the *heavy tail property of normalized singular values* of weight matrices factorized using SVD[1]. LRCs that can better express themselves as low-rank pose a heavy-tail distribution of normalized singular

---

[1]Note that WeLore's non-uniform rank selection strategy can be easily adapted to activation-guided SVD techniques (Yuan et al., 2023) and our experiments suggest that our techniques can significantly boost their

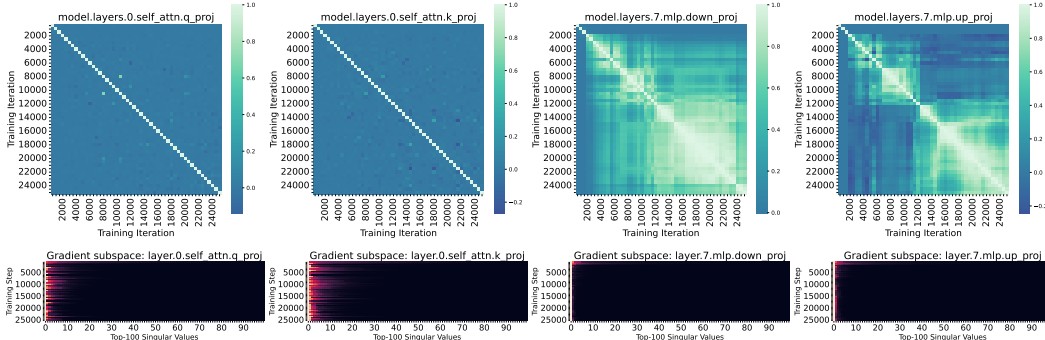

Figure 2: (Row 1) Cosine similarity of the gradients obtained from various checkpoints during pretraining of LLaMA-130M on C4 dataset for 25,000 training steps using Adam Optimizer. Detailed layer-wise cosine similarity is presented in Appendix A.9. (Row 2) Low-rank Gradient Subspace of LLaMa-130M pretraining where each row of individual subplot represents the singular values obtained with SVD over gradient matrices. All gradients are obtained using a fixed batch of data samples for uniformity in results.

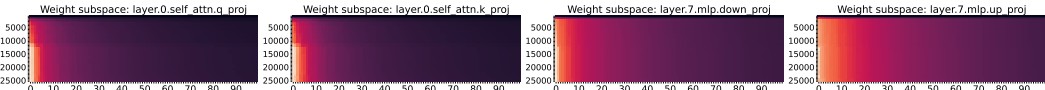

Figure 3: Emergence of Low-rank Weight Subspace during pretraining of LLaMA-130M on C4 dataset for 25,000 training steps using Adam Optimizer. Each row of individual subplot represents the singular values of weights at a given training step for the layers (*e.g.*, mlp.up_proj, attn.q_proj).

values, and are subjected to high-rank reduction without significant loss in information. On the other hand, N-LRCs that do not have low-rank structures well converged can be left either with full rank or undergo minimal rank reduction subjected to target reduction ratio. WeLore reduction ratios can be estimated *using the pre-trained checkpoints* in once-for-all layers fashion without any dependence on downstream or pretraining calibration datasets that makes it easily adaptable across and implementation-friendly with *minimizing sensitivity* to calibration datasets.

The **unique** proposition of WeLore lies beyond a low-rank compression technique, in facilitating memory and parameter-efficient finetuning. WeLore proposes to back-propagate only on significantly compressed LRCs in their low-rank format (eliminating the need to store full-rank optimizer states, full-rank weights & activations in memory) that can *mimic* the optimization similar to full-finetuning (LRCs and N-LRCs jointly). Note that unlike LoRA (Hu et al., 2021) and its variants, which add new low-rank matrices unrelated to the original weight (proxy optimization), we rely on existing low-rank subspaces from pre-trained weights, without introducing additional parameters (instead, reducing parameters) and thereby operating in the original optimization trajectory. Our extensive experiments across continual finetuning with C4 dataset (Figure 1) & downstream task finetuning (Figure 8) illustrate that LRC-based WeLore finetuning can match (even outperform) the performance of full-finetuning with a fraction of trainable parameters, higher throughput, and notably less GPU memory need (*e.g.,* in comparison to full-finetuning 50% low-rank compressed LLaMA-2 7B, WeLore have ∼0.35× trainable parameters, ∼3× better throughput, ∼0.6× GPU requirement).

## 2 LOW-RANK SUBSPACE OF WEIGHTS AS A CONSEQUENT OF GRADIENTS DYNAMICS DURING PRETRAINING

The continuous growth in scale of LLMs is making the computational and memory costs of inference and finetuning them notably prohibitive. Finetuning LLMs has recently been very successful in boosting their capabilities to follow instructions, adopting response-generating style, and limiting undesirable behaviors like hallucination, generating toxic contents, etc. To enable the democratization of these abilities with consumer-grade GPUs, enormous efforts are directed toward LLM compression

---

performance (Table 2). However, we intentionally focus on simple SVD at weight space to overcome the high sensitivity of activation-based SVD on calibration datasets along with facilitating ease in system-level implementation (Chavan et al., 2023).

and parameter-efficient fine-tuning techniques. Among several techniques (*e.g.,* sparsity (Jaiswal et al., 2023b;c; Lee et al., 2019; Frankle & Carbin, 2019; Jaiswal et al., 2023a; Yin et al., 2023; Liu et al., 2023a), quantization (Liu et al., 2023b; Kim et al., 2023; Dettmers et al., 2023a; Frantar et al., 2022; Lin et al., 2023; Dettmers et al., 2023b)), low-rank decomposition of weight matrices draws special attention as compressed linear layers remain fully differentiable and all parameters are trainable while being able to leverage the existing highly efficient kernels over floating point matrices.

Surprisingly, most existing works (Hsu et al., 2022; Kaushal et al., 2023; Li et al., 2023; Wang et al., 2024) primarily focus on developing new algorithms for effectively decomposing the pre-trained weight matrices. Their under-explored assumption revolves around uniform existence of low-rank structures within gigantic matrices in LLMs. In addition, they fail to explore their emergence and variability across different layer types (*eg.,* attention, mlp) and position (*eg.,* middle or terminal layers) within the deep LLM model. Recently, Galore (Zhao et al., 2024) presented a theoretical sketch suggesting gradients during pretraining of LLMs exhibit low-rank behavior but didn't provide details of the dynamics and variability of these low-rank structures across different layers of LLMs. Inspired by GaLore, we aim to explore: ① How does gradient behavior changes during pretraining across different layers of LLMs? ② How does gradient dynamics lead to the emergence of low-rank structure across gradients and weights? ③ Does the low-rank structure uniformly prevalent in the pre-trained weights of LLMs? If not, can we build an adaptive low-rank strategy subjected to quantification meerged low-rank properties during the pretraining?

Firstly, Figure 2 (row 1) represents the pairwise cosine similarity of the gradients captured (using a fixed batch of data) from model checkpoints of LLaMa-130M sampled every 500 training steps during pretraining from scratch on C4 dataset. The first two subplots of row 1 indicate the gradient behavior of `self_attn.q_proj` & `self_attn.k_proj` from the 1st transformer block while the next two subplots are for `mlp.down_proj` & `mlp.up_proj` from the middle 7th transformer block of 11 block deep LLaMa-130M model. Figure 2 (row 2) presents the corresponding gradient subspace of these layers where every row of each subplot indicates the singular values obtained by SVD decomposition of gradient matrices during pretraining iterations. Our observations can be summarized as:

- Gradient dynamics is *not uniform* across all the sub-layers of the LLMs during pretraining.

- Gradients behavior across some layers (*e.g.,* majority of middle mlp layers) illustrate an *early-bird saturation* property and can't accumulate rich error signals from the training dataset during pretraining.

- To some layers (*e.g.,* attention matrices from the terminal transformer blocks) the behavior is opposite and where *gradient behavior keeps changing continuously* throughout pretraining.

- Connecting previous observations with the gradient subspace in row 2, we found a strong correlation in the emergence of low-rank structure (heavy-tail illustrated as bright colors to the left) as a direct consequence of continuously changing rich error propagation signals.

Next, we attempt to understand how these observations translate to the emergence of low-rank structures in the weight matrices of the model. Figure 3 presents the corresponding emergence of weight low-rank structures throughout pretraining within layers. Our findings are summarized as:

- We found the emergence of low-rank structure across the weight matrices very early during pretraining which becomes explicit and notable as pretraining progresses. Similar to gradient subspace, we found that not all layers can express themselves as low-rank and this property significantly varies subject to position (middle layers or terminal layers) and role (attention layers or mlp layers).

- We found a strong correlation between the gradient dynamics and the low-rank emergence across the weight matrices (*e.g.,* early gradients dynamics saturation of `model.layer.7.mlp.down_proj` leading to non-low-rank gradient subspace which ultimately reflects in the weight matrix not converging to low-rank[2]). As a consequence of

---

[2]A sharp bright line across the subplots in Figure 3 to the left suggests heavy-tail distribution of singular values. A heavy-tail singular value distribution from SVD is a favorable property that indicates the matrix can be well compressed using a few singular values without introducing large reconstruction errors.

the cumulative accumulation of gradients within a low-rank gradient subspace, the corresponding weight matrices of the layers exhibit the emergence of high-quality stable low-rank subspace.

# 3 WeLore: Adaptive Low-Rank Weight Projection of Pretrained Weights

LLMs are omnipresent and recently the race of scaling them have attributed to gigantic computational and memory footprints. Among numerous efforts towards democratization for consumer-grade GPUs, low-rank decomposition of pretrained weights as a product of two smaller dense matrices receives special attention because it can leverage the highly optimized floating-point dense matrix multiplication kernels unlike sparsity and quantization which require specialized kernels to be written, often different for each hardware backend in order to enable speedup. Recently, several works (Hsu et al., 2022; Kaushal et al., 2023; Yuan et al., 2023; Wang et al., 2023; Saha et al., 2023; Wang et al., 2024) have explored matrix factorization of LLMs' pretrained weights. We found that these works primarily focus on improving SVD using more informative signals like activation, fisher information and applying it unilaterally (same rank reduction ratio) across all the weights. As discussed in previous section, low-rank emergence varies significantly across candidate weights in a pretrained checkpoint. To this end, we pose an under-explored question: *How can we carefully curate a layer-adaptive rank reduction ratios for all layers in the pretrained checkpoint?*

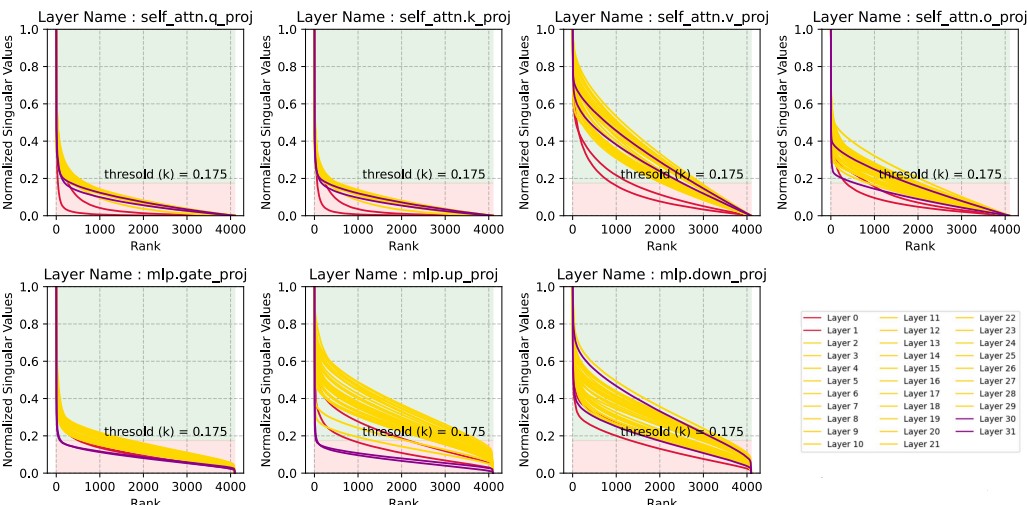

Figure 4: Normalized singular values of the weight matrices corresponding to different layers of LLaMa-2 7B pretrained checkpoint. Each subplot indicate sorted & normalized 4096 singular values corresponding to different layers (*e.g.,* `self_attn.q_proj`) from 32 transformer blocks.

Figure 4 presents the normalized 4096 singular values corresponding to different layers across 32 transformer blocks of LLaMa-2 7B. It can be clearly observed that for some layers (*e.g.,* `self_attn.q_proj`, `self_attn.k_proj`, `mlp.gate_proj`) elicit a heavy tail behaviour indicating better low-rank expressivity compared to others (*e.g.,* `self_attn.v_proj`, `mlp.down_proj`). Another important observation to note is that majority of the layers from the front and tail blocks of the model tend to have better low-rank property which aligns with our gradient behavior study. Heavy tail indicates only a small fraction of singular values carries maximum information and the corresponding matrix can be well approximated using a fraction of basis vectors from SVD with marginal reconstruction error.

Weighted Low-rank Projection (WeLore) proposes a data-agnostic and implementation-friendly normalized singular value thresholding technique[3] with only one global hyperparameter (`k`) as shown as the shaded red and green region in Figure 4 for layer-adaptive rank reduction. More specifically, we

---

[3]Normalization helps us to compare singular value distribution across all layers at the same scale.

aim to preserve normalized singular values greater than the threshold `k` shown as shaded green region. For a given effective rank reduction ratio[4] of $ERR$, the global threshold `k` can be approximated using linear search[5] over `np.linspace(0, 1, 0.005)` with condition as follows:

$$\frac{\sum_l \texttt{sum}(\mathcal{S}_{W_l} < \texttt{k})}{\sum_l \texttt{len}(\mathcal{S}_{W_l})} \approx ERR \tag{1}$$

where $W_l$ represents the weight matrix of layer $l$ and $\mathcal{S}_{W_l}$ is the array of sorted normalized singular values estimated with `torch.svd`$(W_l)$. Note that `k` estimation is not computationally expensive as the $\mathcal{S}_{W_l} \forall l$ can be calculated before searching for `k`. Given a weight matrix $W_l^{4096 \times 4096}$ and $\mathcal{S}_{W_l} = \{s_1, s_2, ..., s_{4096}\}$, the compressed rank $r$ can be provided as $r$ = `np.sum`$(\mathcal{S}_{W_l} \geq$ `k`$)$. In compressed format, $W_l^{4096 \times 4096}$ can be represented as a composition of two small matrices $A_l^{4096 \times r}$ and $B_l^{r \times 4096}$ where $r << 4096$. As it can be read from the Figure 4, for `k = 0.175` which indicate an aggregated 50% rank reduction, majority of the `self_attn.q_proj` from 32 transformer blocks of LLaMa-7B can undergo significant reduction $\geq 90\%$ (*i.e.*, $r < 400$). On the other hand, layers such as `self_attn.v_proj` & `mlp.down_proj` which are not low-rank friendly will receive high $r$.

Given $r_l$ for all the layers $l$ in the pretrained checkpoint, WeLore categorizes all the layers into two broad categories - Low-rank Components (LRCs) and Non-Low-rank Components (N-LRCs). Layers with heavy-tail which can be effectively represented with $r_l < 0.5 \times$ `rank`$(W_l)$ falls in LRCs while the rest falls in N-LRCs. We replace weight matrices of all LRCs in pretrained checkpoint as composition of two small matrices $A$ & $B$ to achieve notable parameter reduction (*e.g.*, $\times 0.67$ parameters with $\mathbf{R} = 0.5$) saving memory and compute during inference and fine-tuning (low-rank weight representation allows gradients and optimizer states to be in low-rank during finetuning).

## 4 MEMORY-EFFICIENT LOW-RANK AMICABLE FINETUNING

Parameter-Efficient finetuning techniques (PEFT) which enable LLMs to perform a new task with minimal updates has received enormous attention to their ability to allow fine-tuned by only updating a small number parameters. Unlike LoRA and its varients which finetune a *small added fraction* of parameters to original pretrained weight checkpoints not relevant to original pretraining optimization, WeLore provides an alternative approach by capitalizing the gradient perspective to *select a small fraction of weights* from the pretrained model which can undergo fine-tuning. As discussed above, LRCs exhibits low-rank structure with rich gradient dynamics while N-LRCs can't be well-expressed in low-rank format. To this end, WeLore make the following proposal:

Given a low-rank compressed checkpoint with LRCs and N-LRCs, finetuning with backpropagation **only through LRCs** (frozen N-LRCs) can closely mimic the performance of full-finetuning (sometimes better) with considerable memory and compute reduction. Given that LRCs are represented in low-rank format, both gradients and optimizer state will by default in low-rank saving finetuning cost.

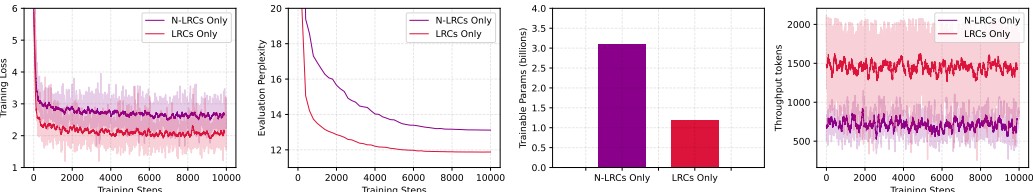

Figure 5: Finetuning statistics and performance comparison of Low Rank Components (LRCs) and Non-Low-Rank Components (N-LRCs) layers of a 50% compressed LLaMa-2 7B model with C4. Note that all finetuning hyperparameters are kept same in both settings for fair comparison.

**Empirical evidence that LRCs are better at learning than N-LRCs:** Here, we investigate the relative difference in performance and compute expenses related to finetuning LLMs. Figure 5

---

[4]Effective Rank Reduction Ratio (ERR): $1 - \frac{\sum_l rank(W_l^{Compressed})}{\sum_l rank(W_l^{Original})}$

[5]Pseudo-code for `k` estimation is provided in Appendix A.3. We also provide pre-estimated values for LLaMa-7B and LLaMa-13B used in the submission in the Appendix A.5.

present our comparison of continual finetuning statistics of LLaMa-7B pretrained checkpoint with 50% effective rank reduction ratio on C4 dataset for 10,000 training steps. Red color indicate finetuning by back-propagating only through LRCs (freezing all the N-LRCs) while magenta color indicate finetuning N-LRCs (freezing LRCs). It can be clearly observed that despite $\sim 3\times$ more trainable parameters, training loss as well as the validation perplexity of finetuning N-LRCs are significantly *under-performing* in comparison to finetuning LRCs. Moreover, it is important to note that the throughput achieved by LRCs is $\sim 2\times$ in comparison to N-LRCs which can be attributed to the parameter-efficient low-rank represented weight matrices, gradients, and optimizer state.

## 5 EXPERIMENTS AND ANALYSIS

In this section, we first investigate the superiority of WeLore's layer-adaptive rank reduction ratio for effective low-rank compression of pre-trained checkpoints of LLMs. Next, we investigate the effectiveness of WeLore for joint compression and LRCs-focused parameter efficient finetuning performance across several downstream tasks. We additionally report the empirical GPU requirements for performing inference and finetuning across different compression ratios. Our extensive experiments illustrate that unlike prior works which *either focus on low-rank compression or parameter-efficient finetuning*, WeLore **uniquely** differentiates itself by proposing an effective low-rank compression strategy and presents a novel angle of memory and parameter-efficient fine-tuning using LRCs for comparable performance to full-finetuning.

| Rank Reduction | LLaMa2-7B [PPL: 7.03] | | | | LLaMa2-13B [PPL: 6.53] | | | |
|---|---|---|---|---|---|---|---|---|
| | Uniform Reduction | OWL Reduction | WeLore Reduction | WeLore Finetuned | Uniform Reduction | OWL Reduction | WeLore Reduction | WeLore Finetuned |
| 10% | 10.58 | 12.11 | 7.13 | 7.15 | 7.17 | 7.2 | 6.55 | 6.55 |
| 20% | 16.43 | 14.49 | 8.28 | 7.40 | 8.61 | 8.53 | 6.96 | 6.68 |
| 30% | 91.99 | NaN | 14.41 | 8.18 | 13.99 | 11.63 | 8.66 | 7.42 |
| 40% | NaN | NaN | 78.17 | 9.47 | 1178.03 | 56.06 | 24.92 | 8.69 |
| 50% | NaN | NaN | 1836.62 | 11.87 | 4167.79 | 7984.39 | 1142.53 | 11.40 |

Table 1: Perplexity comparison of LLaMa-7B with various rank reduction techniques at different reduction ratios. Gray column indicates the performance after memory-efficient continual finetuning of LRCs on 1×A6000 GPU using C4 dataset (7M tokens) with token `seqlen` of 1024.

### 5.1 IMPLEMENTATION DETAILS

**Network Architectures:** For understanding gradient dynamics and its consequent on the weight space during pretraining, we adopt the LLaMa-130M architecture following (Lialin et al., 2023; Zhao et al., 2024). For our continual and downstream finetuning experiments, we adopted the pretrained checkpoint of LLaMa-2 7B, LLaMa-2 13B and Mistral-7B[6] from HuggingFace.

**Low Rank Compression:** For low-rank compression using WeLore for LLaMa-2 7B and 13B models, we used `torch.svd(W_l)` to decompose a layer $l$'s weight matrix $W_l^{m \times n} = A^{m \times r} B^{r \times n}$ where r is decided by the heavy tail distribution of the singular values of $W$ as described in Section 3. If $W$ belongs to LRCs, it will be replaced with a composition of two linear layers with low-rank matrices $A$ & $B$ to improve the computational efficiency. For baselines, we compared with commonly used uniform rank reduction (Hsu et al., 2022; Kaushal et al., 2023) and adopted recently proposed outlier-weighed non-uniform ratio (OWL) (Yin et al., 2023). We additionally augmented activation-guided SVD techniques (Yuan et al., 2023) with WeLore's adaptive layer-wise rank reduction ratio to understand how it can benefit them.

**Continual and Downstream Finetuning:** For continual finetuning settings, we finetune the WeLore compressed LLaMa-2 7B and 13B models at different compression ratios using C4 dataset. The C4 dataset is a massive collection of Common Crawl's web crawl corpus, meticulously filtered and cleaned to ensure high-quality language modeling and training. For downstream task finetuning of compressed models, we consider a good mixture of tasks from commonsense reasoning and math reasoning, namely `CommonsenseQA`, `BoolQ`, `CoinFlip`, `SVAMP`, `BigBench`, `StrategyQA`. For comparison, we have used two baselines: (i) `LoRA`: LoRA (Hu et al., 2021)

---

[6]Perplexity and Downstream Performance results of Mistral are presented in Appendix A.4.

introduces low-rank adaptors for training the models, $W = W_0 + UV$, where $W_0$ is the pretrained weights, which are frozen during training. In our setting, we associate $U$ and $V$ with all the components of the LRC and N-LRC of the compressed model and fine-tune them while keeping $W_0$ frozen. (ii) `GaLore` (Zhao et al., 2024): GaLore projects the gradient into low-rank format and updates the optimizer states and projects it back for updating weights. In this setting, we perform finetuning of both LRCs and N-LRCs (full-finetuning) with projected low-rank gradients. Our finetuning experiments start from the same checkpoint and hyperparameter settings for fair comparison.

## 5.2 EXPERIMENTAL RESULTS AND ANALYSIS

### 5.2.1 WELORE FOR COMPRESSION OF PRE-TRAINED LLMS

① **WeLore identifies Non-Uniform rank reduction ratio across layers to limit performance drop.**
We investigated the layer-wise rank reduction ratio achieved by WeLore and found it to be highly non-uniform where some layers can be compressed significantly higher than others. In addition, note that layers from the first and last few transformer blocks are compression-friendly. Figure 6 illustrates the rank reduction ratios after 50% effective rank reduction of LLaMa-2 7B pretrained checkpoint using WeLore. Interestingly, it can be noted that `self_attn.q_proj` & `self_attn.k_proj` layers can be expressed as low-rank with $> 90\%$ compression. Moreover, the majority of layers from transformer blocks at the front and tail end are better at compression due to well-converged low-rank properties. The green region indicates LRCs while the red region indicates the N-LRCs components.

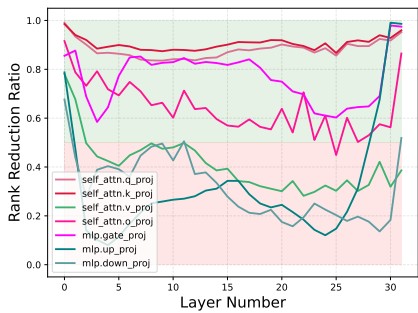

Figure 6: Layer-wise rank reduction ratio of 50% compressed LLaMa-7B.

② **WeLore is superior than Uniform and Outlier-Weighted reduction ratio.** Low-rank decomposition of LLMs has been primarily investigated with unilateral (same rank) reduction across all the weights. In contrast, WeLore presents non-uniform rank reduction ratio guided by emerged low-rank structures in pretrained checkpoints. Table 1 presents the comparison of perplexity of LLaMa-2 7B and 13B models on C4 validation dataset with EER of 10% to 50% when compressed with WeLore and our two baselines. It can be clearly observed that as EER increases, the perplexity of the baseline compressed model significantly explodes (becomes NaN for LLaMa-7B), but WeLore retains the perplexity within a reasonable range. For example, WeLore is $\sim 6.4 \times$ better than 30% Uniform EER for LLaMa-2 7B and $\sim 47 \times$ better than 40% Uniform EER for LLaMa-2 13B. Note that OWL reduction tends to perform sometimes better than Uniform reduction, but its degradation in performance with increasing EER is more severe.

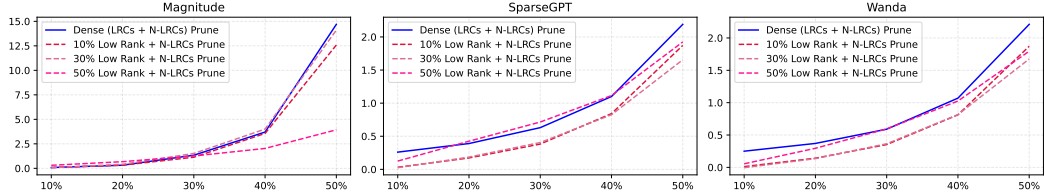

Figure 7: Perplexity comparison (↑) for further compression of N-LRCs using SoTA LLM pruning methods for LLaMa-2 7B on C4. Note that we calculated the increase in perplexity wrt. the initial perplexity of dense and low-rank compressed checkpoints with ERR of $r\%$.

③ **Investigating further Compression Opportunity with SoTA LLM Pruning.** Recently (Yin et al., 2023) investigated the activation outlier-based non-uniform sparsity ratios for different transformer blocks within LLMs. A careful observation of their layer-wise sparsity ratio reveals that the majority of middle transformer blocks can be subjected to a higher pruning ratio which is **complementary** to WeLore low-rank reduction ratio that favours terminal blocks being low-rank friendly. We therefore ask an unexplored question: *How does LLM performance changes when we further compress only the dense N-LRCs using SoTA pruning methods?*

| Reduction | Total Params | Model Memory | seqlen = 512 | seqlen = 1024 | seqlen = 2048 | seqlen = 4096 |
|---|---|---|---|---|---|---|
| 0% | 6738.42M | 13,579 MB | 14,467 MB | 15,145 MB | 17,193 MB | 24,519 MB |
| 30% | 5794.25M | 11,993 MB | 12,565 MB | 12,923 MB | 14,549 MB | 20,853 MB |
| 50% | 4543.67M | 9,501 MB | 10,125 MB | 10,433 MB | 12,049 MB | 18,377 MB |
| 70% | 3072.84M | 6,657 MB | 7,285 MB | 7,625 MB | 9,233 MB | 15,549 MB |

Table 3: Empirical estimate of Inference GPU Memory Requirement (measured with GPUtil library) of LLaMa-2 7B compressed with WeLore with varying context sequence length.

Figure 7 presents the increase in the perplexity of LLaMa-2 7B on the C4 dataset when we compress a dense checkpoint (blue) using SoTA LLM pruning methods. We compared it with further compressing dense N-LRCs of WeLore checkpoints with ERR of 10%, 30%, and 50%. Our key observations are: (i) WeLore checkpoints can further enjoy high compression with sparsification of dense N-LRCs without signification performance drop to a noticeable sparsity ratio (*e.g.,* WeLore checkpoint with ERR of 50% can be additionally sparsified using Wanda (Sun et al., 2023) with < 2 points increase in perplexity); (ii) ad-hoc sparsification of LRCs and N-LRCs (dense) suffers higher performance degradation compared to N-LRCs which demands actively exploring amalgamation of different compression techniques for LLMs to ripe maximum benefits; (iii) development of better sparsity algorithms (*e.g.,* Wanda (Sun et al., 2023), SparseGPT (Frantar & Alistarh, 2023)) clearly retain their benefits even in mixed compression settings.

⑤ **WeLore's Non-uniform Ratios also benefits Activation-Guided Rank Decomposition.**

Activation-guided SVD techniques (Yuan et al., 2023; Wang et al., 2024) have been found more effective than weight-oriented SVD methods by managing activation outliers and adjusting the weight matrix based on the activation distribution. Despite our work focusing on simple weight SVD to enable easy adaptation and minimize sensitivity to calibration datasets, we conducted experiments to illustrate that WeLore can also significantly benefit from Activation-SVD. Table 2 and Appendix A.2 present the perplexity comparison of Uniform ActSVD wrt. when it is augmented with the non-Uniform reduction ratio identified by WeLore.

| Model | | LLaMa2-7B | |
|---|---|---|---|
| Rank Reduction | Uniform Reduction | Uniform+ActSVD Reduction | WeLore+ActSVD Reduction |
| 10% | 10.58 | 7.24 | 7.05 |
| 20% | 16.43 | 7.75 | 7.21 |
| 30% | 91.99 | 8.85 | 7.87 |
| 40% | NaN | 11.33 | 9.75 |
| 50% | NaN | 17.03 | 14.76 |

Table 2: Performance benefit (PPL on C4) of WeLore reduction ratio on ActSVD.

⑥ **Inference Memory Statistics of WeLore Compression.** In this section, we investigate the memory requirement for inference with WeLore compressed models. Table 3 how WeLore allows reducing the memory requirement to load the model parameters by substituting the full-rank weight matrices in their low-rank format. Given a consumer-grade GPU like GeForce RTX 4090, WeLore can facilitate inference with 4K context length where the original model will flag an OOM error.

| | | LLaMa2-7B [1×] | | | | LLaMa2-13B [1×] | | |
|---|---|---|---|---|---|---|---|---|
| Reduction → | 30% | 50% | 60% | 70% | 30% | 50% | 60% | 70% |
| Compressed Params | 0.85× | 0.67× | 0.56× | 0.45× | 0.83× | 0.64× | 0.53× | 0.43× |
| LoRA Finetuning GPU Requirement | 8.21 26,859MB | 12.48 25,129 MB | 21.23 24,621 MB | 382.24 23,711 MB | 7.49 46,162 MB | 21.53 42,293 MB | 27.99 41,191 MB | 124.44 40,513 MB |
| Galore Finetuning GPU Requirement | 9.02 29,773 MB | 18.57 25,673 MB | 396.05 24,155 MB | 670.29 22,777 MB | 8.02 54,378 MB | 60.07 45,810 MB | 2454.03 41,703 MB | 3396.19 37,448 MB |
| WeLore Finetuning GPU Requirement | 8.18 30,197 MB | 11.87 28,281 MB | 17.87 27,193 MB | 47.92 25,955 MB | 7.42 52,452 MB | 11.40 47,091 MB | 19.20 43.136 MB | 73.59 42,922 MB |

Table 4: Performance (perplexity) comparison of compressed LLaMa-2 7B & 13B with WeLore adaptive rank selection technique and continual finetuning with LoRA and GaLore wrt. WeLore.

### 5.2.2 WeLore for Joint Compression and Parameter-Efficient Finetuning

① **Continual-Finetuning of WeLore Compressed Models.** In this section, we investigate the performance statistics of LRC-focused WeLore tuning with respect to LoRA and GaLore in different compression ratios. Given a pretrained checkpoint (LLaMa-7B and 13B), we first perform rank reduction using WeLore with varying ERR between 30-70% which can achieve up to 55% reduction in total model parameters. For fair comparison, we perform continual finetuning of the compressed

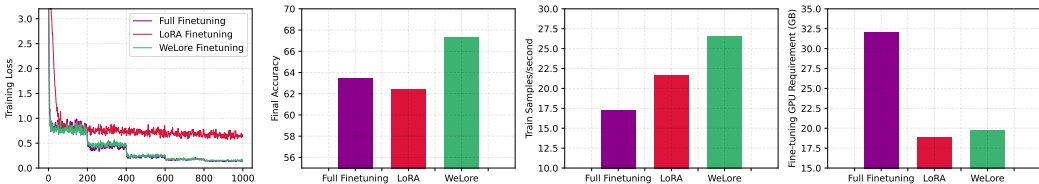

Figure 8: Downstream Finetuning statistics and performance comparison of WeLore vs. full-finetuning and LoRA of a 50% compressed LLaMa-2 7B model with StrategyQA dataset with `max_len` of 512. All finetuning hyperparameters are kept same in all settings for fair comparison.

model using LoRA, GaLore and WeLore with sequence length of 1024 on 0.7M tokens; all other hyperparameters are set identically. Table 4 illustrates the superiority of LRCs-focused WeLore finetuning where the benefits increase with a higher degree of compression.

② **Downstream-Finetuning of WeLore Compressed Models.** To understand the effectiveness of LRCs-only WeLore finetuning, we consider full-parameter finetuning, LoRA, and GaLore for dense pretrained checkpoint as well as WeLore compressed checkpoint of LLaMa-7B. We conducted several experiments across various compression ratios on math and commonsense reasoning tasks and report our performance in Table 5. Surprisingly, LRCs-based finetuning of WeLore compressed models tends to closely match and sometime outperform even the dense as well as compressed full-parameter finetuning of LLaMa-7B pretrained checkpoint. Additionally, the performance achieved by LRCs-focused WeLore finetuning is significantly and consistently higher than both LoRA and GaLore across all the tasks while having memory requirements close to LoRA. Figure 8 illustrate that unlike LoRA, LRC-focused WeLore finetuning can closely mimic the loss trajectory of full-finetuning with significantly low GPU memory requirements and can achieve throughput greater than LoRA based fine-tuning.

| Reduction | Method | CommonsenseQA | SVAMP | BoolQ | CoinFlip | BigBench[7] | StrategyQA |
|---|---|---|---|---|---|---|---|
| | Dense Full Finetune | 77.052 | 40.672 | 88.189 | 75.000 | 83.742 | 69.581 |
| | Dense LoRA Finetune | 76.414 | 50.090 | 70.962 | 69.333 | 80.995 | 68.690 |
| | Dense GaLore Finetune | 75.339 | 41.667 | 68.362 | 65.667 | 77.980 | 67.325 |
| | Full Finetune | 75.925 | 40.667 | 84.005 | 51.333 | 83.364 | 70.783 |
| 30% | LoRA | 64.537 | 44.333 | 81.776 | 61.333 | 68.750 | 65.255 |
| | GaLore | 64.015 | 42.667 | 80.892 | 55.333 | 75.735 | 62.490 |
| | WeLore | 76.744 | 53.333 | 85.040 | 98.667 | 81.818 | 69.648 |
| | Full Finetuning | 71.908 | 38.333 | 83.603 | 49.000 | 90.224 | 68.502 |
| 40% | LoRA | 54.386 | 36.667 | 75.021 | 54.667 | 76.002 | 65.154 |
| | GaLore | 52.078 | 36.333 | 71.039 | 50.333 | 77.910 | 65.440 |
| | WeLore | 76.003 | 42.667 | 81.646 | 98.666 | 87.857 | 67.794 |
| | Full Finetuning | 70.120 | 25.333 | 80.113 | 53.333 | 89.431 | 63.411 |
| 50% | LoRA | 35.382 | 23.667 | 75.482 | 50.667 | 54.022 | 62.408 |
| | GaLore | 35.122 | 21.667 | 71.552 | 47.667 | 58.975 | 61.336 |
| | WeLore | 70.516 | 30.667 | 80.377 | 94.666 | 87.802 | 67.290 |

Table 5: Downstream performance of Dense and WeLore compressed LLaMa-2 7B checkpoint under full-finetuning along with memory-efficient finetuning techniques (LoRA and GaLore). All downstream finetuning is performed starting from the same initial checkpoint state for fair comparison.

## 6 CONCLUSION

We study the emergence of non-uniform low-rank structures across different layers of transformer blocks from gradient behavior perspective. We present WeLore, an adaptive layer-wise low-rank compression strategy for low-rank decomposition which can achieve high compression ratio with minimal drop in performance. The unique proposition of WeLore lies in categorizing weight matrices of pretrained models into two borad categories - LRCs and N-LRCs based on their ability to express themselves as low-rank. We conducted extensive experiments to validate that LRCs pose better trainability than N-LRCs. Given limited compute & memory budget, WeLore recommends finetuning LRCs while keeping N-LRCs frozen with back-propagation for maximal gain (sometimes better than full-finetuning). The primary limitation of our work remains limited exploration for only the LLaMa family of models and unexplored benefits of WeLore for training LLMs from scratch.

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

# A APPENDIX

## A.1 BACKGROUND WORK

**Memory-Efficient Finetuning:** Memory-efficient fine-tuning of LLMs aims to address the significant costs associated with their fine-tuning. This field encompasses several notable techniques. For instance, Prompt Learning Methods optimize input tokens or embedding while keeping the model's remaining parameters static Hambardzumyan et al. (2021); Zhong et al. (2021). Layer-freezing techniques enhance training efficiency by selectively freezing certain layers Liu et al. (2021); Brock et al. (2017); Li et al. (2024). Additionally, Adapter Methods introduce a small, update-focused auxiliary module into the model's architecture, significantly reducing the number of trainable parameters, as introduced by Houlsby et al. (2019); Diao et al. (2022). Among them, one noteworthy technique is Low-Rank Adaptation (LoRA) (Hu et al., 2021) and its successors (Renduchintala et al., 2023; Sheng et al., 2023; Xia et al., 2024; Zhang et al., 2023; Hayou et al., 2024; Hao et al., 2024; Liu et al., 2024), which introduces a low-rank weight adapter for each layer to reduce the memory footprint by only optimizing the adapter. These low-rank adapters can then be seamlessly merged back into the original model.

Unlike LoRA which performs proxy optimization over additional parameters while keeping the original parameters frozen, WeLore backed by an understanding of gradient dynamics suggests finetuning the original parameters of LRCs in represented in low-rank to mimic full-finetuning. Recently, (Biderman et al., 2024) found that full finetuning is more accurate and sample-efficient than LoRA across several task categories and WeLore can be an effective alternative to achieve the benefits of full-finetuning within a limited compute and memory budget.

**Low Rank Compression:** Large Language Models (LLMs) have succeeded remarkably across various natural language processing tasks. However, the massive scale of these models poses significant challenges in terms of storage efficiency and computational complexity. Among several techniques of LLM compression (*e.g.,* pruning, quantization, etc.), low-rank decomposition which retains only the top-k components in the low-rank space have special privilege to leverage the existing highly efficient kernels over floating point matrices. (Hsu et al., 2022) developed a data-aware modification of SVD that incorporates approximate second-order gradient information. Similarly, (Yuan et al., 2023) proposed a data-aware decomposition method that minimizes activation error. One primary drawback of these reductions is that they uniformly reduce rank across all weight matrices. In contrast, our work experimentally validates existence of non-uniform low-rank expressiveness across different layers and should be accounted for during low-rank compression. Recently, (Zhao et al., 2023; Wang et al., 2023) found that dynamic rank selection during pretraining can achieve comparable prediction performance as full-rank counterpart.

## A.2 ACTIVATION BASED SVD

| Model | LLaMa2-chat-7B | | | |
|---|---|---|---|---|
| Rank Reduction | Uniform Reduction | WeLore Reduction | Uniform+ActSVD Reduction | WeLore+ActSVD Reduction |
| 10% | 10.97 | 6.65 | 6.60 | 6.53 |
| 20% | 63.63 | 8.09 | 7.08 | 6.90 |
| 30% | nan | 19.60 | 8.43 | 8.24 |
| 40% | 28027.73 | 254.74 | 12.56 | 10.94 |
| 50% | 22029.66 | 3209.67 | 26.02 | 15.80 |

Table 6: Perplexity of Wikitext-2 under comparison of LLaMa-V2-chat with various rank reduction techniques at different reduction ratios. The gray column highlights the use of activation-based SVD.

## A.3 ADAPTIVE THRESHOLD SELECTION

---

**Algorithm 1:** Adaptive Threshold Selection Algorithm in WeLore

---

**Input:** A LLM with weights $\boldsymbol{\theta}$, target reduction ratio $s_p$, current reduction ratio $s_t$, reduction tolerance $s_\delta$, threshold incremental value $H_i$.

**Output:** A compressed model $\boldsymbol{\theta}$ satisfying the target reduction ratio $s_p$, singular threshold $H$

**Initialization:** Initialize a singular threshold threshold $H = 0$

**while** *not* $(s_p + s_\delta > s_t > s_p - s_\delta)$ **do**

    **for** *each MLP layer tensor $\boldsymbol{\theta}^l$ in $\boldsymbol{\theta}$* **do**

        $sv^l \leftarrow \texttt{calculate\_singular\_values}(\boldsymbol{\theta}^l);$

        $sv_n^l \leftarrow \texttt{normalize\_singular\_values}(sv^l);$

        $p^l \leftarrow 0;$ **for** *each $s$ in $sv_n^l$* **do**

            **if** $s < H$ **then**

                $p^l \leftarrow p^l + 1;$

    $P_r \leftarrow \sum_l p^l;$

    $s_t \leftarrow P_r / P_t;$

    **if** $s_p + s_\delta \geq s_t \geq s_p - s_\delta$ **then**

        **break;**

    **else**

        $H \leftarrow H + H_i;$

---

## A.4 GENERALIZATION OF WELORE TO MIXTRAL-7B PRETRAINED CHECKPOINT

|  | 5% | 10% | 20% | 30% | 40% | 50% |
|---|---|---|---|---|---|---|
| Uniform Rank Reduction | 9.67 | 12.31 | 78.695 | 6746.48 | 162301.04 | 248042.97 |
| OWL Rank Reduction | 9.02 | 11.63 | NaN | NaN | NaN | NaN |
| WeLore Rank Reduction | 8.19 | 8.76 | 11.90 | 30.69 | 429.08 | 1351.32 |
| WeLore Finetuned Rank Reduction | 8.18 | 8.32 | 8.92 | 9.71 | 14.85 | 21.37 |

Table 7: Perplexity-based performance comparison of WeLore Adaptive Rank reduction.

|  | CommonsenseQA | SVAMP | BoolQ | StrategyQA |
|---|---|---|---|---|
| Full Finetuning | 68.45 | 19.66 | 75.09 | 62.37 |
| LoRA | 68.03 | 20.22 | 73.97 | 61.43 |
| GaLore | 65.77 | 12.68 | 73.12 | 61.08 |
| WeLore | 69.36 | 21.59 | 77.41 | 65.17 |

Table 8: Downstream performance comparison of WeLore w.r.t. LoRA, GaLore and Full finetuning.

From Table 7, it can be observed that WeLore generalizes well to Mistral-7B significantly reducing the perplexity of the compressed model in comparison with Uniform rank reduction as well as Outlier-based rank reduction. Moreover, with fine tuning only 20% of parameters (at 50% rank reduction ratio) of 7B model, WeLore can notably outperform LoRA, GaLore as well as full-finetuning for downstream task.

## A.5 PRE-ESTIMATED SINGULAR VALUE THRESHOLDS (κ) FOR LLAMA-2 7B AND 13B

| Model | 10% | 20% | 30% | 40% | 50% | 60% | 70% |
|---|---|---|---|---|---|---|---|
| LLaMa-2 7B | 0.065 | 0.084 | 0.115 | 0.145 | 0.175 | 0.215 | 0.260 |
| LLaMa-2 13B | 0.065 | 0.085 | 0.115 | 0.140 | 0.180 | 0.225 | 0.270 |

Table 9: Thresolds used for low-rank decomposition to different compression level in our experiments for LLaMa-2 7B and 13B. The singular values are calculated using pytorch `torch.svd()` function.

## A.6 WELORE RANK REDUCTION RATIO AND PARAMETER COUNT

| Rank Reduction | LRCs/Trainable # Param Count | N-LRCs/Frozen # Param Count | Total # Model Param |
|---|---|---|---|
| 0% | 0 | 6738.42M | 6738.42M |
| 10% | 291.93M | 6408.90M | 6700.83M |
| 20% | 1225.96M | 5171.58M | 6397.54M |
| 30% | 1450.00M | 4344.25M | 5794.25M |
| 40% | 1498.39M | 3663.72M | 5162.12M |
| 50% | 1453.52M | 3090.15M | 4543.67M |

Table 10: WeLore rank reduction and estimate of total number of LRCs and N-LRCs parameters in the compressed checkpoint.

## A.7 HYPERPARAMETERS FOR CONTINUAL FINETUNING OF LLAMA-7B AND 13B

| Hyperparamter | LLaMa-2 7B | LLaMa-2 13B |
|---|---|---|
| Model Link | Download | Download |
| Batch Size | 1 | 1 |
| Max. Sequence Length | 1024 | 1024 |
| Learning Rate | 5e-05 | 5e-05 |
| Schedular | cosine | cosine |
| Num. Training STeps | 10000 | 10000 |
| Warmup Steps | 500 | 500 |
| dtype | bfloat16 | bfloat16 |

Table 11: Primary hyperparamter configuration setting for continual finetuning of LLaMa-7B & 13B.

## A.8 HYPERPARAMETERS FOR DOWNSTREAM FINETUNING WITH WELORE

| Hyperparameter | CommonsenseQA | SVAMP | BoolQ | CoinFlip | BigBench | StrategyQA |
|---|---|---|---|---|---|---|
| Train Samples (avg. words) | 9741(28.00) | 700 (31.83) | 9427 (14.81) | 350 (37.05) | 295 (34.90) | 1603 (9.61) |
| Test Samples (avg. words) | 1221(27.75) | 300(31.56) | 3270 (14.70) | 150 (36.96) | 74 (35.58) | 687 (9.57) |
| Batch Size | 8 | 8 | 8 | 8 | 8 | 8 |
| Max_length | 512 | 512 | 512 | 512 | 512 | 512 |
| Training Steps | 5000 | 500 | 2000 | 500 | 1000 | 1000 |
| Learning Rate | 0.0001 | 0.0001 | 0.0001 | 0.0001 | 0.0001 | 0.0001 |

Table 12: Hyperparamters settings for downstream finetuning of LLaMa-7B

## A.9 DETAILED QUANTATIVE AVERAGE OF LAYER-WISE COSINE SIMILARITY OF GRADIENTS

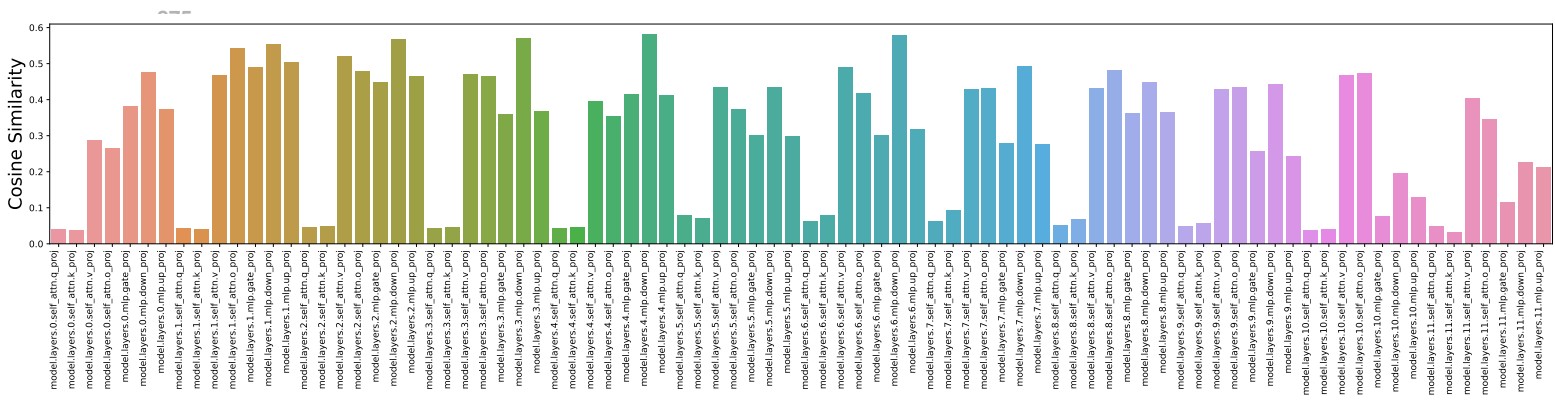

Figure 9: Cosine similarity for gradients of different layers obtained from various checkpoints during pretraining of LLaMA-130M on C4 dataset for 25,000 training steps using Adam Optimizer.

