# OpenReview forum: "From GaLore to WeLore: How Low-Rank Weights Non-uniformly Emerge from Low-Rank Gradients"
_ICLR.cc/2025/Conference — Submitted to ICLR 2025_

### Official Review · Reviewer_9G7z · 2024-10-29

**Soundness:** 2
**Presentation:** 3
**Contribution:** 2
**Rating:** 5
**Confidence:** 4

**Summary:**

This paper establishes a connection between gradient dynamics during pretraining and weight spaces to understand the degree of low rankness of different layer weights. This can be useful for assigning ranks to each layer's weights in an adaptive and more informed manner. The authors unify the goal of memory-efficient finetuning and weight compression by showing that layers with consistent low-rank gradients also result in low-rank weights (termed LRCs in the paper). The experiments show that training only the LRCs can be competitive or even outperform full finetuning for the Llama family of models.

**Strengths:**

1. This paper establishes a link between the gradient dynamics during pretraining and the rank of the weight matrix for a pre-trained model. This relationship leads to adaptive low-rank weights across layers and memory-efficient finetuning.
2. Extensive experiments show that starting with a 50% compressed model, Welore achieves better performance than LoRA, GaLore and full finetuning.

**Weaknesses:**

1. The new observation in this paper i.e. link between low-rank structure within gradient during pretraining and the low-rankness of resulting weight matrices is interesting but it is difficult to understand its significance for adaptive low-rank estimation across layers. WeLore uses a normalized singular value thresholding technique to decide layer-wise ranks, but it's unclear why one would need access to pretraining information (gradient dynamics) for this. Singular Value Decomposition (SVD) on a weight matrix can directly reflect the singular value distribution, which can then be compared across layers. It makes sense to compress the layers with a heavy tail distribution more, and I am not sure one needs access to gradient dynamics to arrive at that conjecture and establish LRCs and N-LRCs.

2.  In Section 4, the authors establish that finetuning LRCs offers memory efficiency because the gradients are inherently low rank for LRCs. However, it is not well concluded as to why freezing N-LRCs during backpropagation closely mimics the performance of full finetuning and even outperforms it. Deciding to train only LRC components offers memory-efficient finetuning but does not explain the performance being akin to full finetuning. The results in Figures 1 and 8 show full finetuning with a 50% compressed model, which in my understanding does not use WeLore style adaptive rank selection. If this is true, then it is expected that such a model has higher perplexity than the WeLore model even before finetuning, and further finetuning a better model (i.e. WeLore) would make it even better. It would be interesting to see results with actual full-finetuning and understand the performance difference between that and Welore.

3. The results presented in 5.2.2 (Table 4) are somewhat confusing. Llama-2 7B has a perplexity of about ~7 on the C4 dataset, but most numbers in this table show extreme degradation compared to that (for example, almost 100% increase in perplexity at 50% reduction with Welore, and worse for higher compression). Table 4 shows a perplexity comparison between LoRA and Welore-based continual finetuning, but Welore tends to require higher GPU memory which defeats the purpose of efficient finetuning. If I try to compare iso-memory, ideally I should be comparing LoRA 1st column (8.21 PPL) with WeLore 3rd column (17.87 PPL) at ~27,000 MB memory.

**Questions:**

1. Point 5 in Section 5.2.1 is not well explained. How does the non-uniform rank reduction ratio identified through WeLore translate to SVD on activation spaces? Does this originate from the connection between activation and weight spaces through gradient dynamics? Further reasoning for this transferability would be well appreciated.

2. In Section 5.1, the authors mentioned that they use Llama-130M to understand the relationship between gradient dynamics and its impact on the weight space of the pre-trained model. Does this property hold across a family of models, or does one always need access to pretraining information to understand which weights are LRCs or N-LRCs?

3. There are some phrases/terms in the paper which are ambiguous. For example, on page 4 the authors use "early-bird saturation property and can’t accumulate rich error signals". Are the authors linking the rank dynamics of layers with their importance during pretraining?

4. Why are all the full finetuning/LoRA results on a 50% compressed model? It is important to understand how Welore performs in context to the standard LoRA setting, where Wo is the full rank frozen weight matrix.

---

> ### Author Response · Authors · 2024-11-25
> **Author Response to Reviewer 9G7z**
>
> We would first like to thank you for your time to review our work. We would now like to address your weakness and questions one by one as follows:
>
> > **Weakness 1**
>
> The primary purpose of our gradient dynamics study is to establish and explain the non-uniform emergence of low-rank properties across different LLM weight matrices as an artifact of non-uniform gradient dynamics. Note that, ignoring the initialization of weight matrices, the resultant weight matrix at the end of pre-training is essentially a prolonged accumulation of gradients, which explains the necessity of non-uniform rank decomposition. We agree with you that we do not need pre-training information for deciding layer-wise rank in WeLore and it can be identified by singular value thresholding. Our gradient dynamics study reveals an significantly important property of LRCs that they are not only very compressible but also very good at fine-tuning on downstream tasks. In our work, WeLore’s two primary contributions of compression and memory-efficient finetuning receive its interpretability from the gradient dynamics (e.g., why some layers are low-rank?, why LRCs are more tunable?). Note that the observations we have from pre-training a small scale LLaMa-130M translate very well for pre-trained checkpoints of LLaMa-7B, and thereby make WeLore generalizable.
>
> > **Weakness 2 and Question 4**
>
> We would like to clarify your weakness part-by-part and support it with some additional experiments. `(1)` Based on the gradient dynamics, we found that the gradients of the N-LRCs are primarily saturated and don't offer informative gradient signals from data for corresponding weight matrix update. This provide an opportunity to trim down N-LRCs during fine-tuning to avoid non-informative and noisy gradient signals, and still be able to match the fine-tuning. `(2)` For Figure 1 and 8, we have indeed used the WeLore style adaptive and non-uniform rank reduction technique to first create a 50% compressed model, and then perform full-finetuning and WeLore-style N-LRCs finetuning keeping exactly the same hyperparameter settings for fair comparison. `(3)` As recommended by you, we provide additional experiments (setting - full LLaMa-7B model with full-finetuning and WeLore style fine-tune where we backpropagate only layers where >90% rank reduction is possible with <1e-3 reconstruction error). We can clearly observe that with only ~27% of the total layers, we can closely match/outperform the performance of full-finetuning and outperform LoRA. Note that this experiment doesn’t perform any compression on the layers of LLaMa-7B but instead only selectively finetune some layers.
>
> | | CommonsenseQA | SVAMP | StrategyQA |
> | ------------- |:-------------:|:-------------:|:-------------:|
> |Full finetuning |77.052 |40.672| 69.581 |
> |LoRA finetuning| 76.414| 50.090| 68.690 |
> |WeLore finetuning| 76.920| 52.667 | 69.325|
>
> > **Weakness 3**
>
> Thank you for raising your doubt. *Firstly,* we agree with you that the degradation at aggressive effective rank reduction of 50% is high (7.03 -> 11.87), but we would like to highlight that it is notably lower than SoTA rank reduction techniques (e.g., SVD-LLM which achieve 12.42 and ActSVD which achieve 26.39 perplexity on C4 validation set despite using calibration dataset) by adopting a non-uniform rank reduction strategy. *Secondly,* regarding your doubt of comparing LoRA 1st column (8.21 PPL) with WeLore 3rd column (17.87 PPL) at ~27,000 MB memory; we would like to highlight that WeLore contribution is two-sided - a joint compression and memory efficient fine-tuning technique. We believe that your argument of comparing a 30% compressed finetuning with LoRA with 60% compressed finetuning with WeLore is not fair from the compression angle. A 30% compressed LoRA finetuned model will have higher computation tax during inference wrt. 50% compressed WeLore. While we agree that WeLore requires comparatively higher memory than LoRA during fine-tuning, it is significantly lower than full-finetuning. Our goal of comparison is centered around within a reasonable memory requirement, how well WeLore based finetuning of x% compressed model compare with LoRA.
>
> > **Question 1**
>
> WeLore allows determination of non-uniform rank reduction ratio for different weight matrices within LLMs. These ratios can be used as a prior while performing activation aware low-rank decomposition methods which take into account activation outliers while weight matrix decomposition. Our experiments (Table 2) found significant performance benefits when adopting WeLore non-uniform rank ratio for this technique. While we agree that activations are indeed connected to weight matrices (which are accumulation of the gradient dynamics over the course of pretraining), we require more clarification from the reviewer about the kind of reading expected for the transferability to better respond to this question.

---

> > ### Author Response · Authors · 2024-11-25
> > **Author Response to Reviewer 9G7z (2)**
> >
> > > **Question 2**
> >
> > Thank you for this great question. We have conducted new experiment on a new model family (Mistral) to answer the question. Our experiments illustrate that WeLore property holds true of other family of model too and it is not always required to access to pretraining information to understand which weights are LRCs or N-LRCs. Our experimental results are as follows:
> >
> > **(A) Perplexity based evaluation:** Dense Mistral-7B Model Perplexity on C4: **8.14**
> >
> > | | 5% | 10% | 20% | 30% |  40% | 50% |
> > | ------------- |:-------------:|:-------------:|:-------------:|:-------------:|-------------:|-------------:|
> > |**Uniform Rank Reduction**| 9.67 |12.31|78.695|6746.48|162301.04|248042.97|
> > |**OWL Rank Reduction**| 9.02|11.63|NaN|NaN|NaN|NaN|
> > |**WeLore Rank Reduction**| 8.19|8.76|11.90|30.69|429.08|1351.32|
> > |**WeLore Finetuned Rank Reduction**| 8.18|8.32|8.92|9.71|14.85|21.37|
> >
> > **(B) Downstream task Evaluation** Performance comparison on Downstream tasks of Mistral-7B with 50% Rank Reduction:
> >
> > | |CommonsenseQA | SVAMP | BoolQ | StrategyQA |
> > | ------------- |:-------------:|:-------------:|:-------------:|:-------------:|
> > | Full Finetuning|68.45|19.66|75.09|62.37|
> > |LoRA|68.03|20.22|73.97|61.43|
> > |GaLore|65.77|12.68|73.12|61.08|
> > |WeLore|69.36|21.59|77.41|65.17|
> >
> > Clearly, it can be observed that WeLore generalizes well to Mistral-7B significantly reducing the perplexity of the compressed model in comparison with Uniform rank reduction as well as Outlier-based rank reduction. Moreover, with fine tuning only ~20% of parameters of 7B model, WeLore can notably outperform LoRA, GaLore as well as full-finetuning.
> >
> > > **Question 3**
> >
> > Thank you for raising this interesting point. Yes, we observe that gradients of some layers saturate during training quite early (we call this as early-bird saturation) and don't accumulate diverse gradients from the training data. To further address your question about directly linking with their importance, we additionally conducted experiments to freeze such layers (average 0.9 cosine similarity of gradients over 100 iterations of pre-training) after their gradients saturate during the rest of pretraining. Very interestingly, we found that the final performance of LLaMa-130M doesn’t significantly varies (freezed validation PPL = 24.623 v/s no-freezing validation PPL = 24.942) which is a indication that these layers holds marginal importance after gradient saturation during rest of the pre-training duration.

---

> > > ### Author Response · Authors · 2024-11-26
> > > **Looking for forward feedback**
> > >
> > > Dear Reviewer 9G7z:
> > >
> > > We really appreciate your insightful and constructive comments that help us improve this paper. We have taken our maximum effort to address all your concerns.
> > >
> > > As the ICLR public discussion phase concludes in one days, we kindly ask if our responses have addressed your concerns. If there are any remaining issues, we’d be happy to provide further clarifications. If you feel that your concerns have been resolved, we would greatly appreciate it if you could consider raising the score.
> > >
> > > Thank you once again for your time and thoughtful input. We hope you have a wonderful day!
> > >
> > > Best wishes,
> > >
> > > The Authors

---

> ### Author Response · Authors · 2024-11-27
> **Looking for forward feedback (2)**
>
> Dear Reviewer 9G7z:
>
> Given that there is only few hours left before the pdf update window closes, we would deeply appreciate response from you for our rebuttal. If you feel that your concerns have been resolved, we would greatly appreciate it if you could consider raising the score. If you have any further question, we would love to address them within the limited time window.
>
> Thank you once again for your time. We hope you have a wonderful day!
>
> Best wishes,
>
> The Authors

---

### Official Review · Reviewer_uJ8A · 2024-11-03

**Soundness:** 3
**Presentation:** 2
**Contribution:** 2
**Rating:** 3
**Confidence:** 4

**Summary:**

The authors introduce a novel non-uniform matrix-decomposition-based model compression method and a continual memory-efficient fine-tuning method based on the compressed model. They propose that the non-uniform distribution of singular values in gradients leads to a corresponding non-uniform low-rank structure in the weights.  The method categorizes weight matrices into low-rank components and non-low-rank components, and sets the low-rank ones as trainable. The authors validate their theoretical findings through extensive empirical evaluations across multiple datasets and model architectures, showing that the proposed method outperforms the uniform and OWL-based decomposition methods approaches in terms of both computational efficiency and model performance.

**Strengths:**

- A novel and effective non-uniform rank determination method is introduced, leveraging straightforward linear search, followed by the proposal of an efficient low-rank decomposition technique.
- The informative weights are effectively identified based on the proposed method, as evidenced by the decomposition and fine-tuning experiments.
- The core of non-uniform decomposition is elucidated through the lens of gradient dynamics, thereby enhancing interpretability.

**Weaknesses:**

- This paper explores the emergence from gradient dynamics to rank determination, yet it lacks a clear linkage between gradients and ranks. If I understand it correctly, the rank determination function does not rely on any gradient information, as indicated by the equal 1.
- Employing the same SVD decomposition method, the main experiment, as depicted in Table 1, omits a SOTA decomposition method baseline, such as SVD-LLM. The current baseline of uniform and OWL-based methods alone is inadequate for a comprehensive effectiveness evaluation.
- The proposed fine-tuning method is not a general fine-tuning method, relying on the proposed decomposition for informative matrix identification. The correlation between the two proposed methods appears somewhat rigid.  The fine-tuning experiments can serve as a dual verification for the informative matrix identified by the previous method, but not a sole fine-tuning method.
- Certain figures lack clarity, for instance, the similar colors employed in Figure 7.

**Questions:**

1. What is the primary challenge addressed in this paper: rank determination or the relationship between gradient dynamics and rank determination?
2. This paper claims that “first study the emergence of low-rank structures across matrices within different layers”, yet, to my knowledge,  there has already been some research about non-uniform rank allocation research, such as :
    - [Feature-based Low-Rank Compression of Large Language Models via Bayesian Optimization]
    - [MoDeGPT: Modular Decomposition for Large Language Model Compression]
- Could the proposed fine-tuning method be applied to fine-tune uncompressed models or models compressed by other methods? If not, it could still serve as a post-calibration method, which remains valuable.
- It would be even better if there were more in-depth research and description on gradients and compression.

---

> ### Author Response · Authors · 2024-11-25
> **Author Response to Reviewer uJ8A**
>
> We would first like to thank you for your time to review our work. We next want to address your weakness and questions one-by-one as follows:
>
> > **Weakness 1: Lacking a clear linkage between gradients and ranks.**
>
> Thank you for raising this point. Our work empirically establishes that low-rank properties across gradients are not uniform across all the components of LLMs. Through our experiments on pre-training a small scale LLaMa-130M, we showed that gradients of some components saturate early during training and don't change over time. To clarify the clear linkage between gradients and rank of a layer, we state that: *Given that we ignore the initialization of weights, weights are inherently a result of cumulative addition of gradients over the course of pretraining.* Layers with early saturation of gradients, didn’t receive enough diverse gradients updates to develop high-quality low rank properties during pre-training. On the other hand, layers which in turn receive diverse low-rank gradients for a prolonged period tend to converge with low-rank compression properties. We will further elucidate this in our revised draft.
>
>
> > **Weakness 2: Comparison with SVD-LLM baseline.**
>
> Thank you for your suggestion. In the following table, we provide comparison with SVD-LLM for perplexity on the C4 dataset on LaMMa-7B. Note that, SVD-LLM is a uniform rank reduction strategy that relies on calibration data for compression, on the other hand WeLore doesn’t require any calibration data for simplicity, speed and eliminating the high sensitivity of activation-based SVD on calibration datasets.
>
> Under fair settings, WeLore non-uniform reduction ratio can significantly benefit from incorporating the usage of calibration data as shown in the following table. This illustrate the significant usefulness of a non-uniform rank reduction ratio strategy provided by WeLore, which can benefit majority of the SoTA novel calibration-dependent and calibration-independent SVD techniques.
>
> | | SVD-LLM (Calibration C4 Data) | WeLore Non-Uniform Ratio | WeLore Non-uniform Ratio + Calibration Data |
> | ------------- |:-------------:|:-------------:|:-------------:|
> |10%|9.32|7.13|7.05|
> |20%|15.84|8.28|7.21|
> |30%|25.11|14.41|7.87|
> |40%|49.83|78.17|9.75|
>
>
> > **Weakness 3/Question 3: Proposed fine-tuning method is not a general fine-tuning method. Could the proposed fine-tuning method be applied to fine-tune uncompressed models or models compressed by other methods?**
>
> We politely disagree with your point. WeLore fine-tuning strategy asserts that, in case of limited memory resources, fine-tuning by backpropagating only though the layers whose gradients behavior haven’t saturated (i.e., LRCs layers) can be highly fruitful and closely mimic full-finetuning. We empirically illustrated that layers which are more low-rank friendly are also better tunable due to their continuously changing gradient dynamics. The proposed fine-tuning can also be used independent of compression and we have conducted additional experiments as following to validate that WeLore observations can be used in isolation for fine-tuning uncompressed models as well.  In the following experiments, we illustrate that with only finetuning ~27% of layers (WeLore LRCs) we can closely match/outperform the performance of full-finetuning and outperform LoRA. Note that this experiment doesn’t perform any compression on the layers of LLaMa-7B but instead only selectively finetune some layers.
>
> | | CommonsenseQA | SVAMP | StrategyQA |
> | ------------- |:-------------:|:-------------:|:-------------:|
> |Full finetuning |77.052 |40.672| 69.581 |
> |LoRA finetuning| 76.414| 50.090| 68.690 |
> |WeLore finetuning| 76.920| 52.667 | 69.325|
>
>
> > **Question 1: What is the primary challenge addressed in this paper: rank determination or the relationship between gradient dynamics and rank determination?**
>
> Thank you for your question. **Firstly**, WeLore investigates the non-uniform emergence of low-rank properties across the weight matrices of LLMs as a consequence of varying gradient dynamics across them which accumulate over the course of pre-training within weight matrices (note that ignoring the initialization, weights are essentially the cumulation of gradients). **Secondly**, WeLore provides an easy-to-implement and data-agnostic strategy of using singular value based thresholding to determine the non-uniform rank reduction ratio for different layers. **Thirdly**, based on the gradient dynamics of low-rank friendly layers, WeLore provides an extensively experimentally validated fine-tuning strategy for fine-tuning in memory constrained settings where it suggests to back-propagate only through the LRCs layers (layers which are more compressible are also more tunable) for maximum gain. Experiments illustrate that it can closely match and sometimes surprisingly outperform full-finetuning.

---

> > ### Author Response · Authors · 2024-11-25
> > **Author Response to Reviewer uJ8A (2)**
> >
> > > **Question 2: This paper claims that “first study the emergence of low-rank structures across matrices within different layers”, yet, to my knowledge, there has already been some research about non-uniform rank allocation research.**
> >
> > Thank you for raising your concern, and we would like to clear out the confusion. In our submission, we didn’t claim that our work is the first research about non-uniform rank allocation. Instead, if read carefully, by keyword “first”, we simply mean the logical ordering of our experiments. For example, “we `first` study the emergence of low-rank structure, then `second` we propose a non-uniform rank technique, then `third` we propose a memory-efficient way of selective layer finetuning”. In addition, we also want to thank you for suggesting new references and we promise to add them in our revised draft.
> >
> > We hope our rebuttal addresses your questions/weakness. If yes, we would politely request you to consider raising your score.

---

> > > ### Author Response · Authors · 2024-11-26
> > > **Looking for forward feedback**
> > >
> > > Dear Reviewer uJ8A:
> > >
> > > We really appreciate your insightful and constructive comments that help us improve this paper. We have taken our maximum effort to address all your concerns.
> > >
> > > As the ICLR public discussion phase concludes in one days, we kindly ask if our responses have addressed your concerns. If there are any remaining issues, we’d be happy to provide further clarifications. If you feel that your concerns have been resolved, we would greatly appreciate it if you could consider raising the score.
> > >
> > > Thank you once again for your time and thoughtful input. We hope you have a wonderful day!
> > >
> > > Best wishes,
> > >
> > > The Authors

---

> ### Author Response · Authors · 2024-11-27
> **Looking for forward feedback (2)**
>
> Dear Reviewer uJ8A:
>
> Given that there is only few hours left before the pdf update window closes, we would deeply appreciate response from you for our rebuttal. If you feel that your concerns have been resolved, we would greatly appreciate it if you could consider raising the score. If you have any further question, we would love to address them within the limited time window.
>
> Thank you once again for your time. We hope you have a wonderful day!
>
> Best wishes,
>
> The Authors

---

### Official Review · Reviewer_i1U4 · 2024-11-04

**Soundness:** 2
**Presentation:** 3
**Contribution:** 2
**Rating:** 6
**Confidence:** 3

**Summary:**

This paper introduces Weight Low-Rank Projection (WeLore) that captures the heavy tail distribution of singular values to identify the suitable rank reduction ratio for matrices in LLMs and only updates the parts of the model with low rank structures. To be specific, they categorize the matrices into Low-rank Components (LRCs) and Non Low-rank Components (N-LRCs) through the normalized singular value thresholding technique. Their method only backpropagates through LRCs and outperform its full-finetuning with less gpu consumptions.

**Strengths:**

1. The paper is generally well written and easy to read.
2. The idea of analyzing weight low-rank subspace through the lens of gradient behavior and adaptively applying low rank structures to the matrices is new.
3. They empirically show their method outperforms other baseline methods with various LLMs and tasks.

**Weaknesses:**

1. Can the authors clarify if WeLore can be applied during pretraining, or if it is limited to fine-tuning scenarios since this method requires the pretrained checkpoint to choose which matrices can be categorized as LRCs? If it is limited to fine-tuning, how does this compare to methods like Galore that can be used during pretraining?

**Questions:**

1. Please fix some typos. For example, in line 179, meerged -> merged. Also, for section 5.2.1, there are total 6 points but no numbering for 4th one.
2. Why does Welore consume less GPU memory than Galore in Table 4 for LLama2-13B 30%?

---

> ### Author Response · Authors · 2024-11-25
> **Authors Response to Reviewer i1U4**
>
> We would first like to thank you for your time to review our work. We appreciate your comments that our paper is well-written and our analysis is novel. We would like to address your weakness and questions as follows:
>
> > **Weakness 1:**
>
> Thank you for raising this point. While the submission primarily explores WeLore from the fine-training perspective, over the course of rebuttal, we have the opportunity to explore how WeLore can be useful during pre-training. WeLore experiments illustrate that gadients of some layers saturate during training quite early (we call this as early-bird saturation) and don't accumulate diverse gradients from the training data. We explored: *If the gradients of some LLM components are saturated, do we actually need to further spend our computational resources on them during pretraining?* To this end, we additionally conducted experiments to freeze such layers (average 0.9 cosine similarity of gradients over 100 iterations of pre-training) after their gradients saturate during the rest of pretraining. Very interestingly, we found that the final performance of LLaMa-130M doesn’t significantly varies (freezed validation PPL = 24.623 v/s no-freezing validation PPL = 24.942) which is a indication that these layers holds marginal importance after gradient saturation during rest of the pre-training duration. We will indeed provide more experimental results for extension of WeLore in the pre-training setting in our final version.
>
> > **Question 1:**
>
>  Thank you for your recommendation to fix the typos and we will address them.
>
> > **Question 2:**
>
> Thank you for your question. We would like to highlight that Galore performs optimization across all the layers in the 30% compressed checkpoint and requires an optimizer state (although in low-rank format) for each component. On the other hand, WeLore recommends only to fine-tune/backpropagate through only a small fraction of LLM components (which are replaced with two low-rank matrices) while keeping the majority of components frozen during fine-tuing and thereby saving memory requirements. Based on your question, note that WeLore based selective layer fine-tuning strategy can also be adopted for GaLore, and we will conduct additional experiments to validate its effectiveness.

---

### Official Review · Reviewer_npk2 · 2024-11-04

**Soundness:** 3
**Presentation:** 3
**Contribution:** 2
**Rating:** 6
**Confidence:** 3

**Summary:**

This paper starts by empirically studying how low-rank structures emerge "differently" across various layers in LLMs during training. Based on this analysis, the authors introduce WeLore, a unified and data-agnostic weight compression and memory-efficient fine-tuning method, by categorizing the weight matrices into low-rank components (LRCs) & non-low-rank components (N-LRCs). Finally, WeLore outperforms its full finetuning baseline with 3x better throughput and ~0.6x GPU requirement in their Llama-2 7B fine-tuning experiments.

**Strengths:**

I personally enjoyed the analysis in Section 2 most. There are many things to be studied in training dynamics and gradient properties of LLMs, which can be used for improving LLM training, and I found analyses in this paper fall under this category. Indeed, authors design the new algorithm (WeLore) based on their analysis, and validate its effectiveness on various experiments including compressing pre-trained LLMs and parameter-efficient fine-tuning.

**Weaknesses:**

The authors stated that "GaLore theoretically argues that the gradient matrix becomes low-rank during training but does not establish how the gradient behavior accumulates in the weight space". While I appreciate their empirical analyses, having some theoretical argument on their weight space analysis can be helpful. I believe that pre-training experiments with WeLore would make the paper even stronger, I am reluctant to lower my score, as I understand many researchers don't have enough computational resources.

**Questions:**

See above.

---

> ### Author Response · Authors · 2024-11-25
> **Authors Response to Reviewer npk2**
>
> We would first like to thank you for reviewing our work and finding it enjoyable. We indeed agree with your comment about the necessity to carefully study the underlying training dynamics of LLMs and exploiting the observations/clues from them to develop better training recipes, architectures, etc. which can improve the computational and memory bottleneck of LLMs.
>
> Over the course of rebuttal, we have the opportunity to explore how WeLore can be useful during pre-training. WeLore experiments illustrate that gadients of some layers saturate during training quite early (we call this as early-bird saturation) and don't accumulate diverse gradients from the training data. We explored: *If the gradients of some LLM components are saturated, do we actually need to further spend our computational resources on them during pretraining?* To this end, we additionally conducted experiments to freeze such layers (average 0.9 cosine similarity of gradients over 100 iterations of pre-training) after their gradients saturate during the rest of pretraining. Very interestingly, we found that the final performance of LLaMa-130M doesn’t significantly varies (freezed validation PPL = 24.623 v/s no-freezing validation PPL = 24.942) which is a indication that these layers holds marginal importance after gradient saturation during rest of the pre-training duration. We additionally agree with your suggestion to have some theoretical arguments on weight subspace, and this can be a default extension for our submission which we are already considering.

---

### Meta-Review · Area_Chair_b8hH · 2024-12-15

**Metareview:**

This submission introduces WeLore, a framework linking gradient dynamics during pretraining with low-rank properties of weight matrices for adaptive compression and memory-efficient fine-tuning of LLMs. While the concept is interesting and supported by extensive experiments, the core claims lack sufficient justification.

Based on the inputs from the reviewers, my conclusion is that the method’s novelty and practical utility are limited, and substantial revisions are needed to clarify its contributions, experimental design, and proper comparison with prior work as mentioned by the reviewers.

I recommend rejection.

**Additional Comments On Reviewer Discussion:**

The general consensus among reviewers leans toward rejection, and upon going through the paper and all the comments, I am also inclined to reject the paper.

---

### Decision · Program_Chairs · 2025-01-22

Reject